# Urban Network Spatial Connection and Structure in China Based on Railway Passenger Flow Big Data

Minmin Li [1,2], Wenhua Guo [3,4], Renzhong Guo [1,2], Biao He [1,2,*], Zhichao Li [5], Xiaoming Li [1,2], Wenchao Liu [3,4] and Yong Fan [1,2]

1. Key Laboratory of Urban Land Resources Monitoring and Simulation, Ministry of Natural Resources, Shenzhen 518060, China; limm@szu.edu.cn (M.L.); guorz@szu.edu.cn (R.G.); lixming@szu.edu.cn (X.L.); fanyong@szu.edu.cn (Y.F.)
2. Research Institute for Smart Cities, School of Architecture and Urban Planning, Shenzhen University, Shenzhen 518060, China
3. Information Center of Ministry of Natural Resources of the People's Republic of China, Beijing 100036, China; whguo@infomail.mnr.gov.cn (W.G.); wcliu@infomail.mnr.gov.cn (W.L.)
4. Technology Innovation Center of Territory & Spatial Big Data, Ministry of Natural Resources, Beijing 100036, China
5. Key Laboratory of Land Surface Pattern and Simulation, Institute of Geographic Sciences and Natural Resources Research, Chinese Academy of Sciences, Beijing 100101, China; lizc@igsnrr.ac.cn
* Correspondence: hebiao@szu.edu.cn; Tel.: +86-0755-2697-9741

**Abstract:** China's transportation industry has made great achievements in the past 40 years of reform and opening up. At the same time, it has gradually accumulated a series of problems. These problems have led to closer and more complex social and economic connection within and between regions of different scales. The existing research only carries out the characteristic analysis of urban network spatial connection and pattern from a single perspective such as "flow space" theory, spatial interaction model and accessibility method, and fails to accurately describe the complex socio-economic relations between regions. Based on the big data of railway passenger flow, this study selected weighted average travel time, railway network density, and the economic connection model to express the urban network spatial connection and structure of China in 2016 from the perspectives of time, space, and interaction. In 2016, the accessibility, connectivity, and total urban external economic connection of the railway network showed a trend of declining from the east to the west. The top 50 cities ranked by interurban economic connection were all located in the central and eastern regions and showed "diamond shape" distribution characteristics. The four diamond-shaped pairs were Beijing-Tianjin-Hebei, Yangtze River Delta, Pearl River Delta, and Chengyu urban agglomerations. This shape was basically in line with the T-shaped space that has existed for a long time in China's regional development. The accessibility, connectivity, and total external economic connection of national-level urban agglomerations were greater than those of regional-level urban agglomerations, and far greater than those of local-level urban agglomerations. The results showed that there was a mismatch between the layout of the railway network and the population. It will still be necessary to focus on strengthening the construction of transportation infrastructure in urban agglomerations and densely populated areas in the future. This study enriches the "flow space" theory, more fully describes urban network spatial connection and structure in China by considering the three perspectives of time, space, and interaction, and can provides reasonable suggestions for the development of national comprehensive three-dimensional transportation network planning, regional spatial structure optimization, and sustainable development.

**Keywords:** big data of railway passenger flow; weighted average travel time; railway network density; economic connection model; spatial interaction

## 1. Introduction

Over the past 40 years of reform and opening up, China has made remarkable achievements in the development of transportation, and a comprehensive transportation system was initially established [1]. In 2020, it formed a national transportation system with expressways and high-speed railways as the framework. Between them, the high-speed railways total 38,000 km, the expressways total 161,000 km, and both are ranked first in the world [2]. Transportation has effectively promoted the development and protection of territorial space, coordinated development between urban and rural areas, and also optimized productivity distribution, which plays a basic, strategic, and service role in economic and social development [3]. At the same time, the national construction of high-speed transportation facilities has also gradually accumulated a series of problems, such as the mismatch between the transportation layout and the population and urban layout [4], the exacerbation of the vertical imbalance by construction of expressways and high-speed railways [5], and the unreasonable competition in the construction of regional transportation system [6], which led to the social and economic connections among regions becoming closer and complex. Regional differences, comprehensiveness, and spatial interaction have always been the core content of economic geography and regional economics. Based on the "flow space" theory, scholars at home and abroad to carry out related research on the pattern and connection of urban network space using geographical methods such as the spatial interaction model and accessibility model, which can provide theoretical and methodological support for regional economic connection, spatial structure, and sustainable development.

### 1.1. Progress in the "Flow Space" Theory

Spanish sociologist Castells first mentioned the concept of "flow space" in *The informational city*, published in 1989, and detailed the "flow space" theory in the Information Age Trilogy, published in 1996 to 1998 [7,8]. The "flow space" theory was applied by the more famous GaWC and the POLYNET team. Based on the "flow space" theory and methods, the GaWC Group has used advanced producer services (APS) metrics for network analysis in urban regions of the world [9]. POLYNET is an international cooperation project integrating experts and scholars from eight European research institutions, and its research on such subjects as the network structure diversity of multi-center urban agglomeration and the functional zoning of urban areas reflects the superiority of the "flow space" theory and methods [10,11].

With the continuous promotion of "flow space" theory, scholars at home and abroad have expressed increasing interest in spatial connection research at different spatial scales and have achieved many important research results. Among them, research based on the perspectives of traffic flow (air flow, railway flow, road flow), information flow (telephone communication volume, Internet social platform data), capital flow (distribution of financial institutions, network distribution of multinational enterprises or cross-regional enterprises) has gradually become a new hot spot [12,13].

The "flow space" theory is different from the traditional "field space," and network location has replaced geographical location is its main feature. Therefore, the transportation network and information network constitute the spatial network, geographical distance becomes speed and cost, increasing the importance of network location.

### 1.2. Progress in the Spatial Interaction Model

Spatial interaction refers to the actual movement of passengers, goods, or information between the starting and ending points, which is the expression of transportation demand and supply in geographical space [14]. Traditional models describing spatial interaction mainly include the gravity model and the potential model [15–17]. Spatial interaction model studies developed earlier in the West. In the 1880s, British demographer Ernst Georg Ravenstein first used Newton's gravity model in the humanities and social sciences [18]. In 1929, Reilly proposed a retail gravity model that identifies retailers controlling market range boundaries in geographic space [19]. Later, Converse developed Reilly's retail

gravity model and proposed the break-point model [20]. In 1967, Wilson derived the maximum entropy model of the spatial interaction theory, starting from the maximum entropy theory [21]. Based on the previous research, many scholars have also proposed spatial interaction models for different application scenarios, such as the mouth particle model [22] and the neural spatial interaction model [23], which have enriched spatial interaction theory and models.

Exploring spatial interaction is an important path to study regional relations, regional economic connection, and regional spatial structure. With the continuous development of information technology, the research on spatial interaction has discovered a trend of integration with GIS spatial analysis, complex network analysis, and structural equation models. As the global urban network gradually attracts the attention of the academic community, it will become the focus for exploring connections on the vertical level using various attribute data, the network structure, function, and relationships among cities and regions of different scales.

### 1.3. Progress in Accessibility Methods

Accessibility is used to measure the connection capability between nodes in the transportation network. Nonetheless, different scholars have defined the concept of accessibility differently. Hansen was the first to define accessibility in 1959, and in his definition, accessibility is the size of the interaction between nodes in a transportation network [24]. In 1979, Morris believed that accessibility was the means to reach a given activity site from a certain place by way of a specific transportation system [25]. Shen developed an employment opportunity accessibility model of a competitive relationship based on the reachability potential model [26]. Since then, scholars have conducted numerous studies on the accessibility of transportation, mainly adopting various related methods based on the characteristics of geometric and topological transportation networks. The methods for assessing transportation accessibility based on geometric networks include the distance method [27], the cumulative opportunity method [28], and the gravity model method [29]. The methods for transportation accessibility based on topological networks have mainly included the matrix method [30] and the space syntax method [31].

Researchers at home and abroad have expounded a variety of definitions of accessibility in terms of time, space, and sociology, but there is no one recognized definition. In general, accessibility mainly describes the transportation network capacity and evaluates the regional spatial structure and transportation network efficiency using time, distance, and formation.

Generally speaking, the existing research only carries out the characteristic analysis of urban network spatial connection and pattern from a single perspective such as "flow space" theory, spatial interaction model and accessibility method, and fails to accurately describe the complex socio-economic relations between regions. Therefore, this study selected the weighted average travel time, railway network density, and the economic connection model to express the spatial connection and pattern of China's urban network from the perspectives of time, space, and interaction. We then explored the spatial connection pattern of China's urban agglomerations and the correlation of spatial connection pattern and sustainable development. Among them, the weighted average travel time reflects the transportation accessibility, the railway network density reflects the connectivity between regions, and the economic connection model reflects the economic radiation capacity of the central cities to the surrounding areas. This study enriches the "flow space" theory and more fully describes urban network spatial connection and structure in China by considering the three perspectives of time, space, and interaction. This research provides reasonable suggestions for the development of national comprehensive three-dimensional transportation network planning, regional spatial structure optimization, and sustainable development.

The structure of the remainder of this paper is as follows. First, the data sources, analytical framework, and research methods are introduced. Second, the spatial and

temporal pattern of China's urban network is characterized, including spatial accessibility and connectivity. Third, the spatial interaction characteristics of China's urban network are analyzed, including the spatial pattern of the total economic connection and interurban economic connection. Fourth, the characteristics of the spatial connection pattern of China's urban agglomerations, and the correlation of the spatial connection pattern and sustainable development are explored. The last section summarizes the conclusions and provides some suggestions to support national comprehensive three-dimensional transportation network planning, regional spatial structure optimization, and sustainable development based on the research results.

## 2. Materials and Methods

### 2.1. Materials

In China, administrative divisions are divided into the provincial administrative district, the prefecture-level administrative district, the county-level administrative district, and the town-level administrative district. The county-level administrative district includes the city-governed district, county-level city, county, autonomous county, flag, autonomous flag, special zone, and forest area [32]. This study took the county-level administrative district as the basic statistical unit and considered the city-governed districts (generally a prefecture-level city contains several city-governed districts) as a whole. The specific data sources are shown in Table 1, and cover: (1) land area, household registered population, and GDP of the county-level administrative district of China in 2016 obtained from the *China County Statistical Yearbook* and the *China City Statistical Yearbook* [33,34]; and (2) vector data of the railway line network, railway stations, and train frequency, acquired from the website https://www.amap.com/ (accessed on 17 November 2021) and the Railway Customer Service Center of China using web crawler technology. Specifically, land area, population, GDP, and train frequency data were spatialized and divided into each county-level administrative district of China; the railway line network and railway station data underwent error analysis and correction, format conversion, projection conversion, scale consistency and other processing, in preparation for the subsequent county-level data overlay analysis.

**Table 1.** The data base of this study.

| Name | Time | Source | Method |
|---|---|---|---|
| Land area, Population, GDP | 2016 | *China County Statistical Yearbook, China City Statistical Yearbook* | Spatial visualization |
| Railway line network, Railway stations, Train frequency | 2016 | AMAP, Railway Customer Service Center of China | Web crawler technology |

Based on the above data, there were a total of 60,297 railway passenger shifts per day in China and a total of 3293 interurban railway passenger shift pairs per day. Figure 1 reflects the spatial distribution pattern of the train network in China. At the county level, train frequency presented typical regional characteristics: the train frequency in the eastern region was significantly greater than that in the western region, 65.17% of the daily train frequency of contact between counties was 1–10, 20.01% of the daily contact frequency between counties was 10–25, and only 1.52% of the daily contact frequency between counties was more than 100. The areas with more frequent train contact were mainly distributed in the Beijing-Tianjin-Hebei region (such as Beijing-Tianjin), the Yangtze River Delta urban agglomeration (such as Suzhou-Wuxi-Changzhou), and the Pearl River Delta urban agglomeration (such as Shenzhen-Dongguan, Guangzhou-Dongguan).

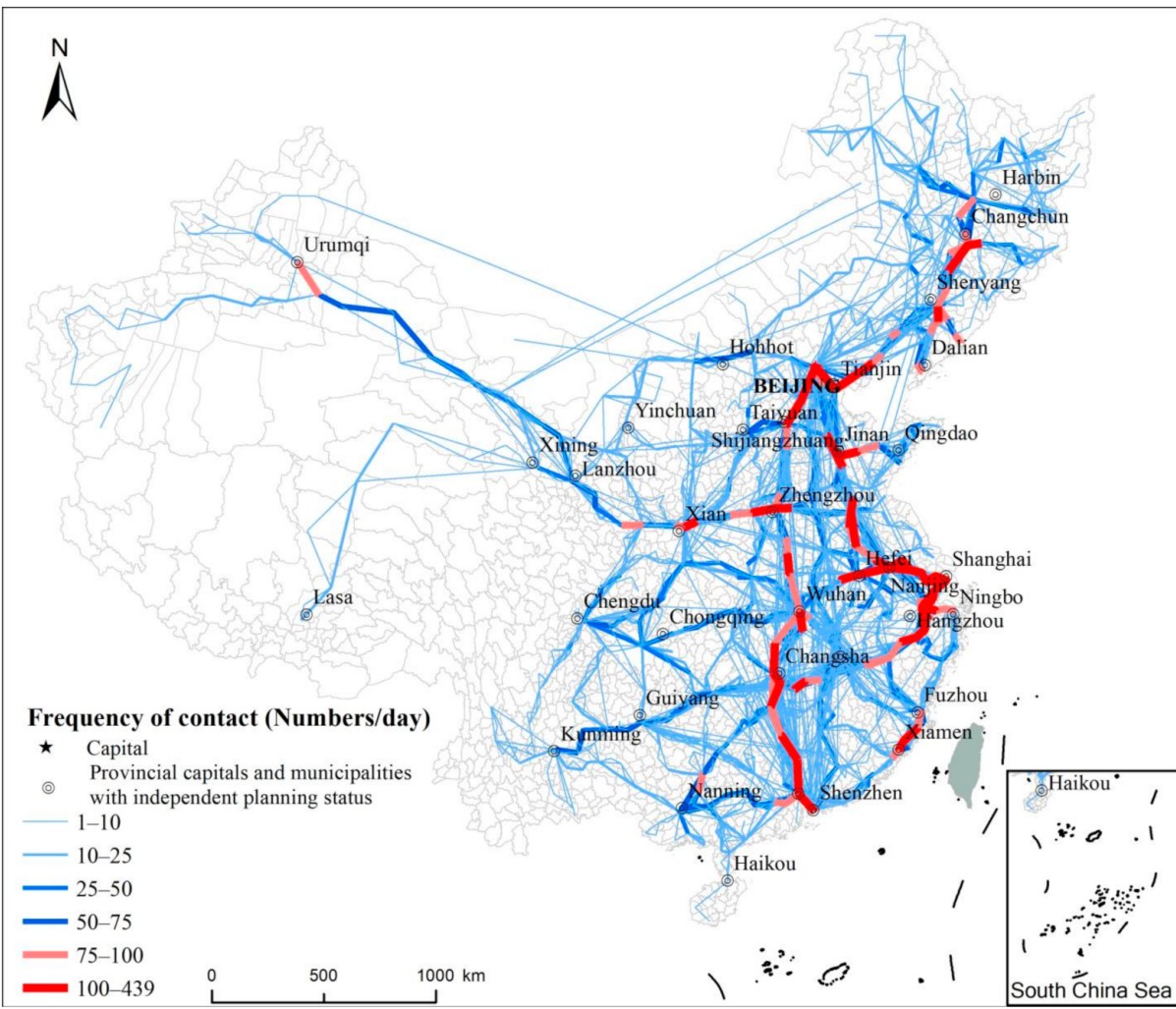

**Figure 1.** Spatial distribution of China's train network in 2016.

*2.2. Methods*

In order to analyze the urban network spatial connection and pattern in China more comprehensively, based on the above theory, methods, and data, the specific technical framework of the study is shown in Figure 2. Specifically, (1) We combed the relevant theories and methods of studying the connection and structure of urban network and selected the theories and models that could be used for this study, such as "flow space" theory, the spatial interaction model, and the accessibility method. (2) Based on pre-processed data, the weighted average travel time that integrated the impact of urban scale and urban development level on accessibility was used to depict the time characteristics of urban network spatial connection and structure, which suggested the time from a certain county to each economic center. Railway network density, population-based railway network density, and economy-based railway network density were used to analyze the space characteristics of the urban network spatial connection and structure, which reflected the connectivity of the interurban network. The economic connection model characterized the interaction of urban network connection and structure that reflected the degree of spatial interaction and economic connection strength between counties. (3) Based on the above methods, we analyzed the spatial and temporal pattern and the spatial interaction of the urban network in China and explored the spatial connection pattern of the urban agglomerations and its correlation to sustainable development.

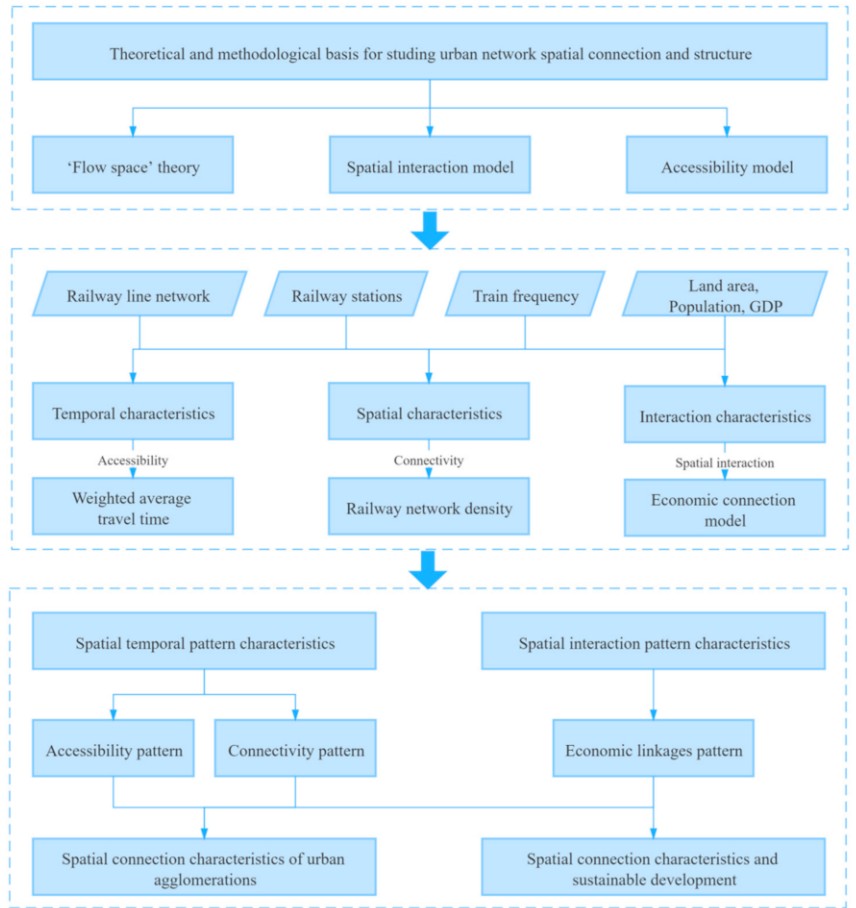

**Figure 2.** Technical framework for this study.

In researching the urban network spatial connection and structure in China based on big data of railway passenger flow, the specific analysis methods were as follows:

### 2.2.1. Weighted Average Travel Time

The weighted average travel time that integrated the impact of urban scale and urban development level on accessibility was used to depict the time characteristics of urban network spatial connection and structure. Weighted average travel time evaluates the proximity of a city to other cities, mainly determined by the urban location, and comprehensively considers the economy, society, and transportation facilities of other cities connected with the city. The smaller the weighted average travel time, the greater the accessibility of the city, and the more convenient the external connection to the city, and vice versa. The model is widely used in transportation accessibility studies [35,36]:

$$A_i = \frac{\sum_{j=1}^{n} (T_{ij} \times M_j)}{\sum_{j=1}^{n} M_j} \tag{1}$$

where $A_i$ indicates the accessibility of county $i$. $T_{ij}$ indicates the shortest railway time from county $i$ to county $j$. $M_j$ is the flow of some socioeconomic element of county $j$, namely, the economic strength of county $j$, and GDP, population, or social commodity sales can be used. Total GDP was used in this study. $n$ is the total number of counties except county $i$.

### 2.2.2. Railway Network Density

Railway network density, population-based railway network density, and economy-based railway network density were used to analyze the space characteristics of the urban

network spatial connection and structure, which reflected the connectivity of the interurban network.

Regional connectivity is reflected by railway network density. The railway network density is a positive index; the greater the value, the denser the railway network, and the better the regional traffic conditions [4]. The railway network density of county $i$ is $C_i$, the railway length of county $i$ is $L_i$, $B_i$ is the land area. The railway network density is then calculated as:

$$C_i = \frac{L_i}{B_i} i = (1, 2, 3 \dots n) \tag{2}$$

To better express the coordination between connectivity, population, and economy, the population-based railway network density $C_{pi}$ and economy-based railway network density $C_{ei}$ are analyzed simultaneously, in which $POP_i$ is the population of county $i$ and $GDP_i$ is the GDP of county $i$:

$$C_{pi} = \frac{L_i}{POP_i B_i} i = (1, 2, 3 \dots n), \tag{3}$$

$$C_{ei} = \frac{L_i}{GDP_i B_i} i = (1, 2, 3 \dots n). \tag{4}$$

### 2.2.3. Economic Connection Model

The economic connection model characterized the interaction of urban network connection and structure that reflected the degree of spatial interaction and economic connection strength between counties.

There are many theoretical models of urban spatial interaction, among them, the Reilly model, Converse model, gravity model, and potential model have great influence and more application [37]. Currently, the gravity model is widely used in studies of regional economics, economic geography, and international trade. In the study of economic geography, the gravity model is generally used to predict the economic connection strength among regions, so it is also called the economic connection model. The economic connection model reflects the radiation capacity of the economic center toward the surrounding areas [18,29]. The specific calculation formula is as follows:

$$L_{ij} = \frac{\sqrt{P_i \times V_i} \times \sqrt{P_j \times V_j}}{T_{ij}^2}. \tag{5}$$

$L_{ij}$ indicates the intensity of economic connection between the two cities; $P_i$ and $P_j$ indicate the total population of county $i$ and county $j$, respectively, and this study used the registered population of the study region; $V_i$ and $V_j$ represent the economic aggregate of county $i$ and county $j$, respectively, and this study used the total amount of GDP of the study region; $T_{ij}$ indicates the shortest railway travel time between the two cities.

The total amount of urban external economic connection indicates the sum of the spatial interaction between each city and all other cities, showing the sum of all the external "flow" of a certain city and reflecting the status of a certain city in the national urban spatial interaction network. The specific formula is as follows:

$$L_i = \sum_{\substack{j = 1 \\ j \neq i}}^{n} L_{ij}. \tag{6}$$

## 3. Results

### *3.1. Spatiotemporal Distribution Pattern of the Urban Network*

### 3.1.1. Railway Network Accessibility in China

The accessibility of a railway network can be expressed by weighted average travel time, as shown in Figure 3. At the national level, the accessibility of a railway network represents the strip distribution and reflects the significant differences between eastern and western regions. The accessibility of the railway network in the eastern region was higher than that of the western region. The weighted average travel time was less than 7 h, which indicated the highest accessibility in these regions and was mainly distributed in the Beijing-Tianjin-Hebei urban agglomeration, Central Plains, Yangtze River Delta, Wuhan metropolitan area, Pearl River Delta and Harbin-Changchun urban agglomeration. This indicated that they had good railway locations and could effectively support regional economic and social development. A weighted average travel time of 7–10 h indicated higher accessibility of the railway network in these regions, which were mainly distributed in the northeast, central, and southeast areas of China. This indicated that they had relatively good railway locations, and that these played a certain role in promoting regional economic and social development. A weighted average travel time of 10–14 h indicated moderate accessibility of the railway network in these regions, which were mainly distributed along the northeast edge and in major western provincial capitals of China. This indicated that the railway locations of these areas were weak and that they had difficulty supporting the economic and social development when only relying on the railway. A weighted average travel time of more than 14 h indicated the lowest accessibility of the railway network in these regions, which was mainly distributed in large western regions and railway-free counties. This indicated that they had the weakest railway locations and that other transportation infrastructure, such as roads and aviation, must compensate for the role of geographical location to promote the coordinated development of these regions.

### 3.1.2. Connectivity of the Railway Network in China

Connectivity of the railway network is expressed through the railway network density, as shown in Figure 4a–c. At the national level, the railway network density presents regional differences. The railway network density in the eastern region was much greater than that in the western region. Areas with great railway network density were mainly concentrated in the urban agglomerations of Beijing-Tianjin-Hebei, Central Plains, Yangtze River Delta, Jianghuai, and the Triangle of Central China. A railway network density of greater than $0.05$ km/km$^2$ reflected the highest connectivity in these regions, which mainly comprised the national railway transportation hub. A railway network density of $0.03$–$0.05$ km/km$^2$ reflected a high connectivity in these regions, which were mainly the important node cities along the railway. A railway network density of $0.01$–$0.03$ km/km$^2$ reflected medium connectivity in these regions, which were mainly the cities with stations along the railway line. A total of 44.79% of China's counties had no railway access, reflecting that these areas relied mainly on other modes of transportation and were mainly distributed in large areas of western China.

The distribution pattern of population-based railway network density and economy-based railway network density differed significantly from the traditional railway network density. The population-based railway network density and economy-based railway network density were the largest in the northeast region, Inner Mongolia, Qinghai, and parts of Tibet in China, while the central and eastern regions with large population density had smaller population-based railway network density and economy-based railway network density. Studies by Li (2018) and others showed that 12% of China's land carries 60% of population and 76% of GDP, and the Beijing-Tianjin-Hebei, Shandong Peninsula, Yangtze River Delta and Central Plains urban agglomerations have the highest population density [38]. The above analysis indicates that there is some mismatch between Chinese railway network layout and population and urban layout. Therefore, we need to improve the railway net-

work construction in central and eastern regions with large population densities, especially Beijing-Tianjin-Hebei, Yangtze River Delta, and other key urban agglomerations.

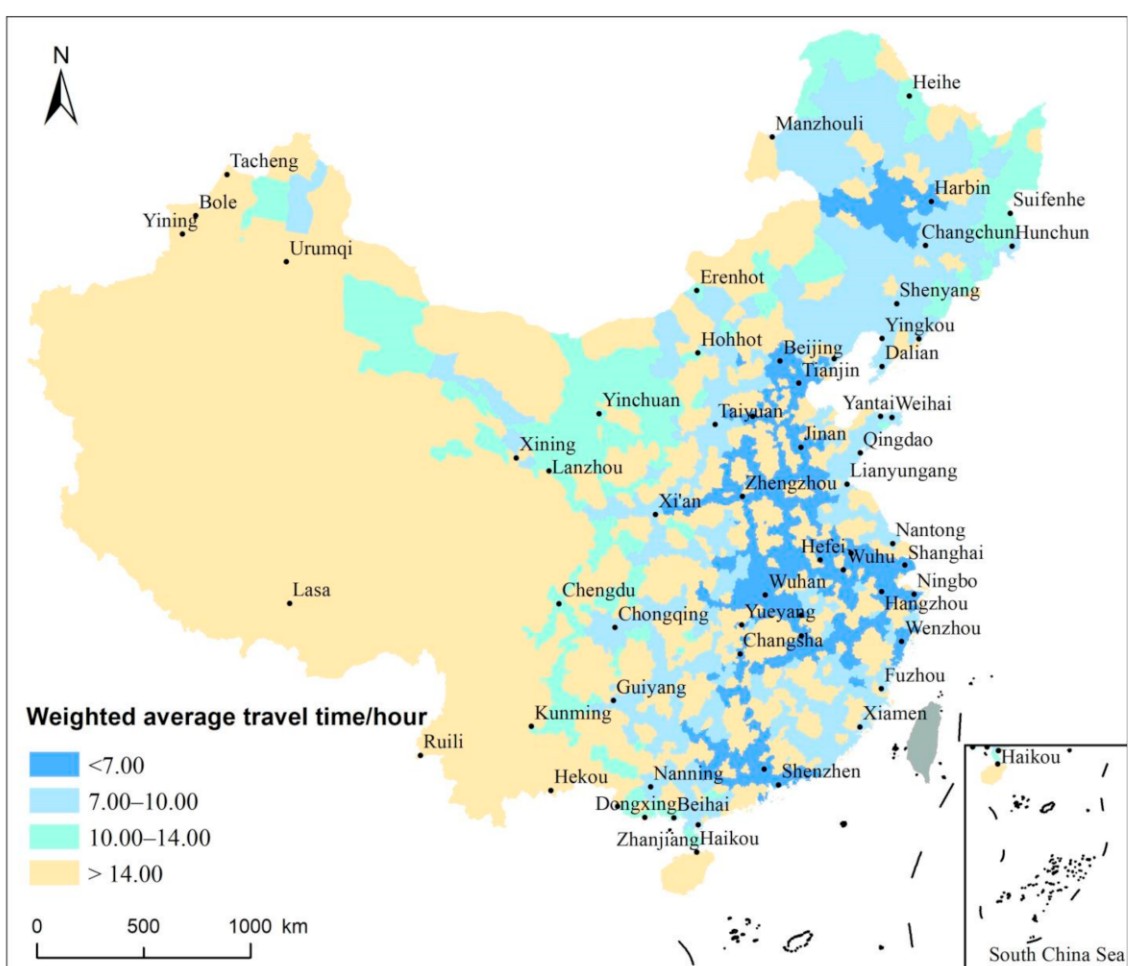

**Figure 3.** Distribution pattern of railway network accessibility.

### 3.2. Spatial Interaction Distribution Pattern of Urban Network

3.2.1. Spatial Distribution Pattern of Total Economic Connection in China

The total amount of urban external economic connection reflects the sum of all the "external flows" of a certain city, which better reflect the status of a certain city in the national urban spatial interaction network. This concept represents the impact of railway construction on the changes of "gravity" in a certain city [22]. The spatial pattern of the total external economic connection in China is shown in Figure 5. At the national level, the total external economic connection in China showed a decreasing trend from the east to the west; the total in the eastern region was significantly higher than that of the central region, and much higher than that of the western region. The cities where the external economic connection was more than 30 (100 million·10 thousand/h$^2$) had the highest status within the national urban spatial interaction network and were mainly the eastern megalopolis and central provincial capitals. The top 10 cities with the highest urban external economic connection were Beijing, Chongqing, Tianjin, Shanghai, Shenzhen, Hangzhou, Guangzhou, Xi'an, Foshan, and Wuhan. The cities where the external economic connection was 20–30 (100 million·10 thousand/h$^2$) had high positions within the national urban spatial interaction network and were mainly distributed around the cities with the highest external economic connection. The cities where the external economic connection was 5–20 (100 million·10 thousand/h$^2$) had a medium status in the national urban spatial interaction network, and they were mainly small and medium-sized cities along the

railway. The cities where the external economic connection linkages were fewer than 5 (100 million·10 thousand/h$^2$) had the lowest status in the national urban spatial interaction network and were mainly distributed in counties with poor economic development along the railway.

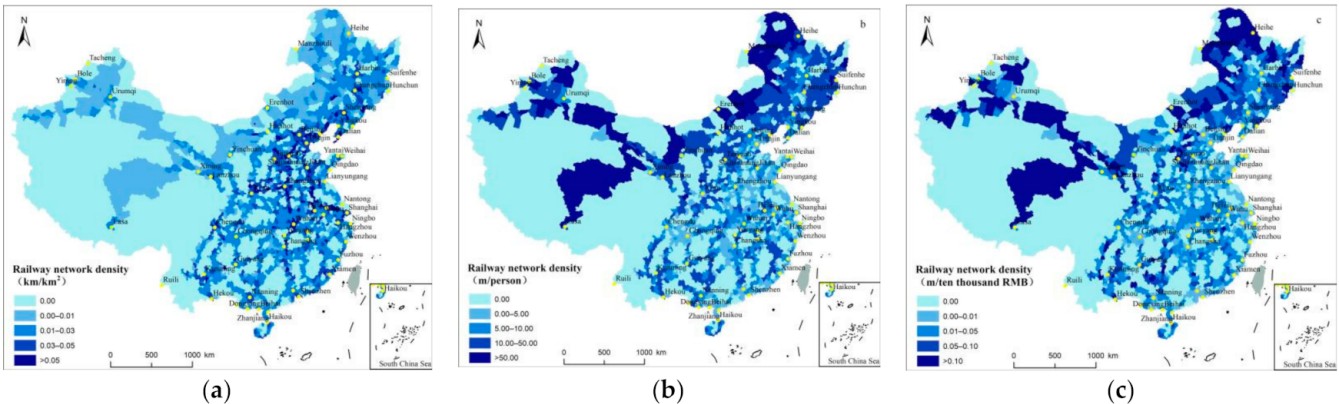

**Figure 4.** Distribution pattern of railway network density. (**a**) Distribution pattern of traditional railway network density; (**b**) distribution pattern of population-based railway network density; (**c**) distribution pattern of economy-based railway network density.

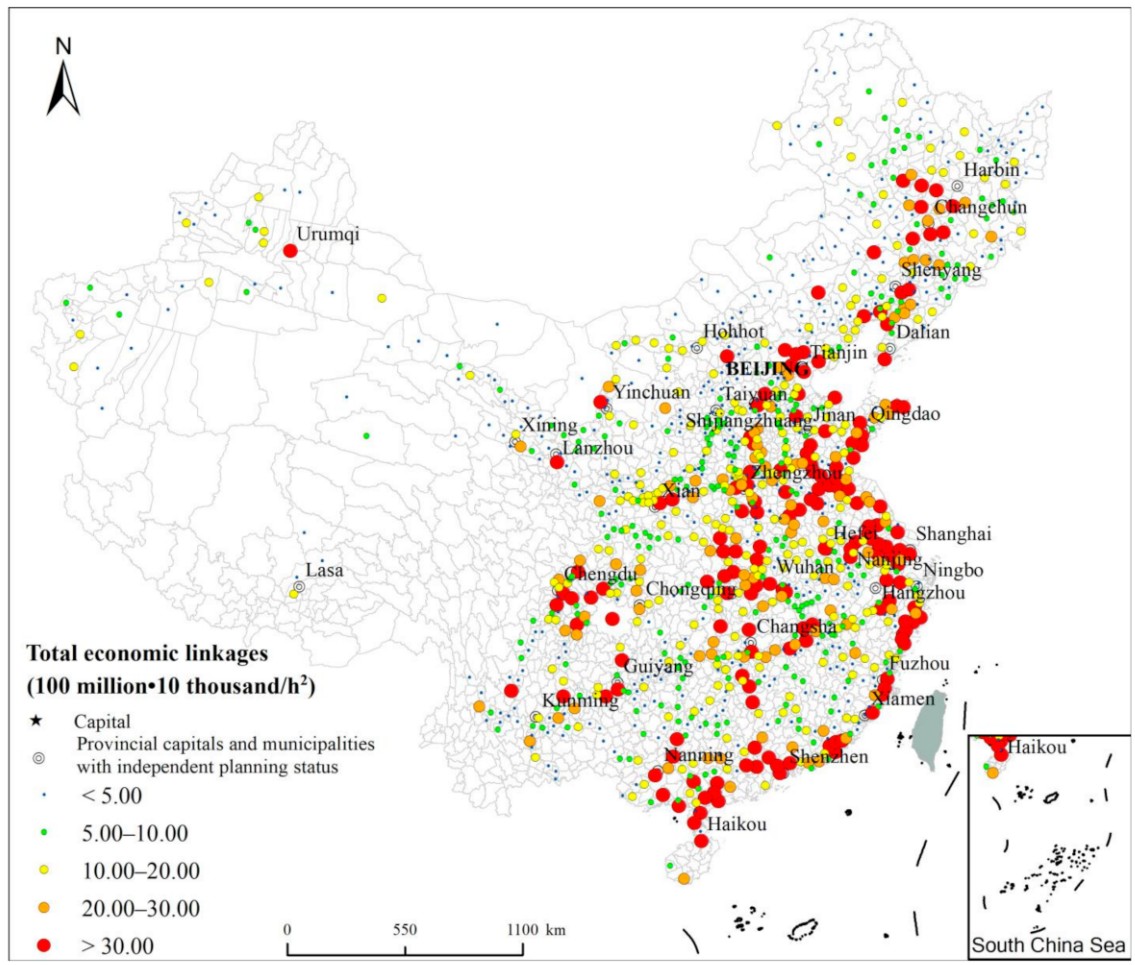

**Figure 5.** Spatial distribution of total economic connection in China.

### 3.2.2. Spatial Distribution of the Top 50 Cities Ranked by Interurban Economic Connection

The interurban economic connection reflects the concept of "flow", which measures the impact of railway construction on the interaction among cities. Spatial distribution of the top 50 cities ranked by interurban economic connection is shown in Figure 6. The top 50 cities ranked by interurban economic connection showed "diamond shape" distribution characteristics. Wuhan was located in the center of the "diamond" distribution. The four diamond-shaped pairs were the Beijing-Tianjin-Hebei, Yangtze River Delta, Pearl River Delta, and Chengyu urban agglomerations. The results indicated that the railway network strengthened the economic connection among the regional central cities. The strongest economic connection between Beijing and Shanghai reflected the close connection between the political and economic centers of China. The top five city pairs ranked by interurban economic connection were Beijing-Shanghai, Chongqing-Shanghai, Beijing-Chongqing, Tianjin-Shanghai, and Shanghai-Guangzhou. Chengdu and Chongqing were the important cities in the western region, and the economic connection between Chongqing and other cities was much greater than Chengdu.

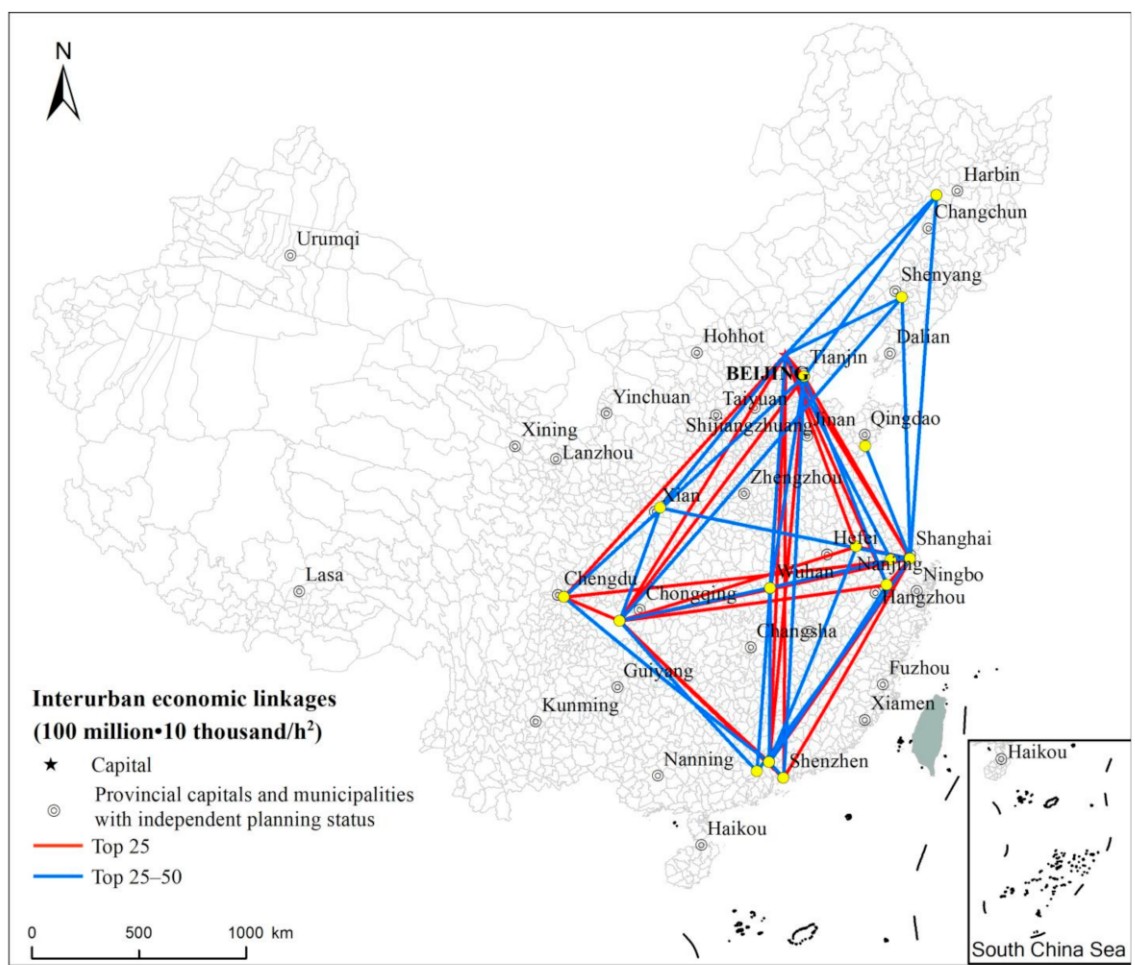

**Figure 6.** Spatial distribution of the top 50 cities ranked by interurban economic connection in China.

Based on mobile signal and OD data from China Unicom in 2018, we characterized the urban network spatial connection and structure using the gravity model [39]. The spatial characteristics also conformed to the "diamond" distribution and were consistent with the conclusion of this study (Figure 7). Therefore, the research results of various data sources have shown that the characteristics of urban network spatial connection and structure in China are not limited by data selection, revealing that China's urbanization has entered the era of urban agglomeration and metropolitan circle. Promoting the development of urban

agglomerations and cultivating modern metropolitan areas have become key tasks in the new type of urbanization in China. In recent years, China has launched the Coordinated Development of the Beijing-Tianjin-Hebei Region, Guangdong-Hong Kong-Macao Greater Bay Area strategy, and the strategy for integrated development of the Yangtze River Delta are all committed to building a world-class urban agglomeration, the Greater Bay area and the metropolitan area, so as to drive high-quality urbanization and participate in international competition and cooperation at a higher level.

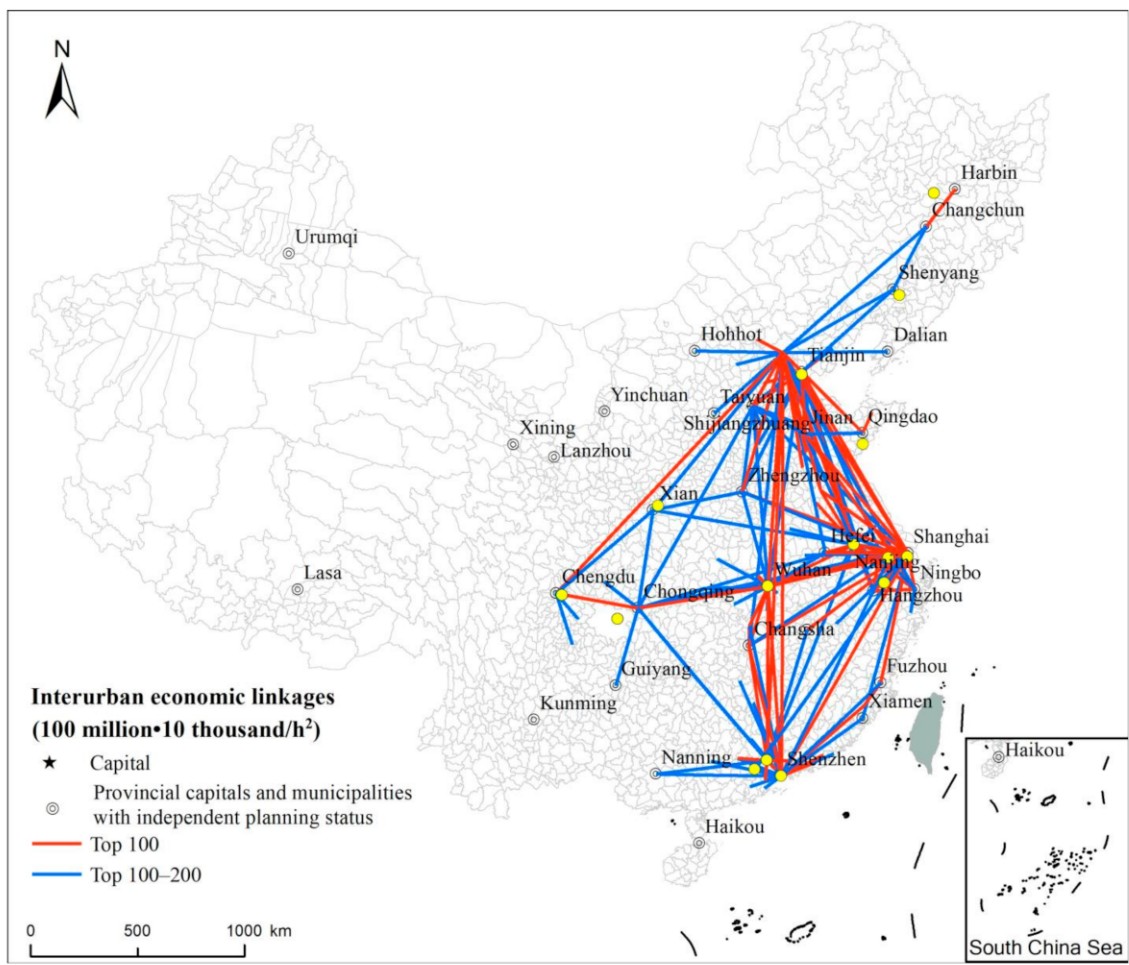

**Figure 7.** Spatial distribution of the top 200 cities ranked by interurban economic gravity.

## 4. Discussion

### 4.1. Spatial Connection and Structure of Urban Agglomerations

There are many distribution structures of urban agglomerations according to different criteria and principles. In China, the main structures of urban agglomerations are Shimou Yao's "6 + 7" plan [40], Chaolin Gu's "3 + 3 + 7 + 17" plan [41], and Chuanglin Fang's "5 + 9 + 6" plan [42]. In this paper, we selected Chuanglin Fang's "5 + 9 + 6" plan to frame the spatial connection and structure in regard to urban agglomeration because this plan considered China's major function-oriented zoning [43] and national urban system planning of 2005–2020 [44] and can describe urban agglomeration comprehensively. The distribution structure of urban agglomeration according to the "5 + 9 + 6" plan is shown in Figure 8, in which pink areas represent five national-level urban agglomerations, yellow areas represent nine regional-level urban agglomerations, and green areas represent six local-level urban agglomerations.

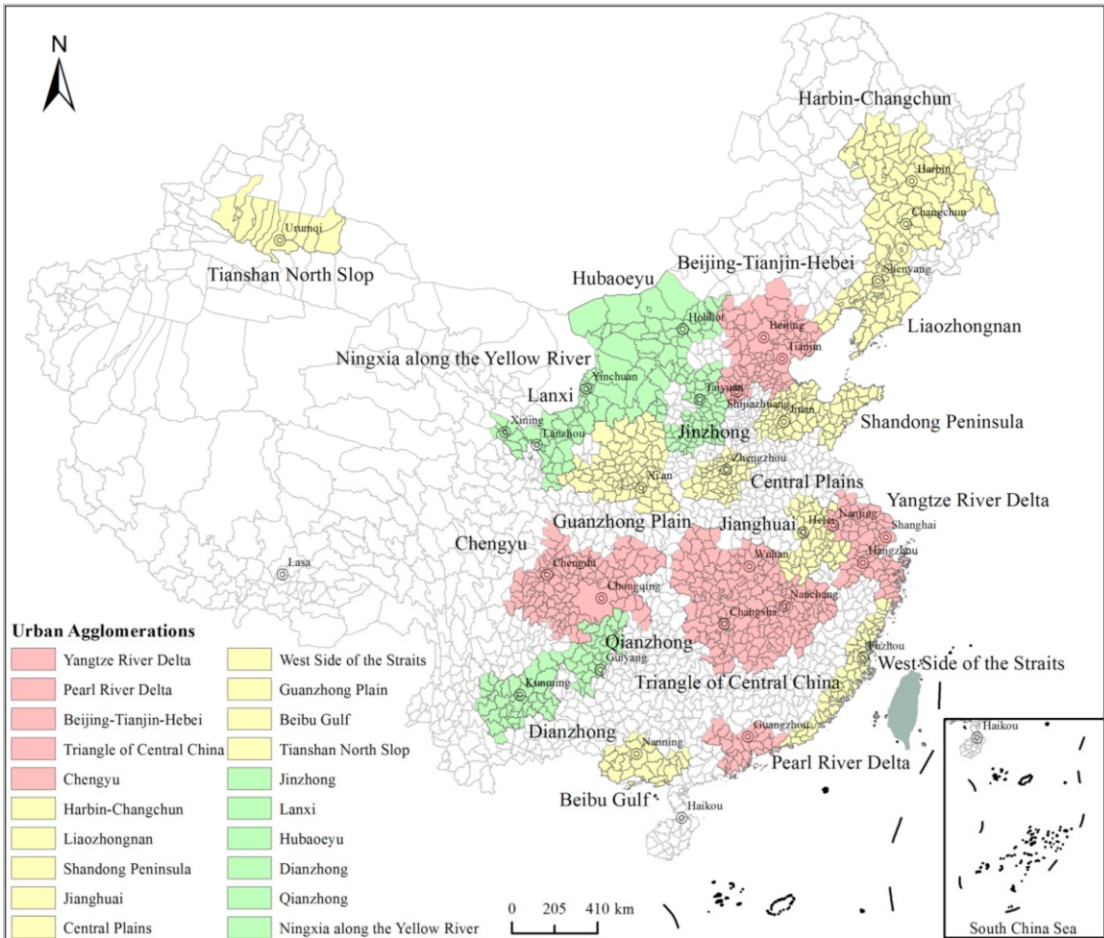

**Figure 8.** Distribution structure of urban agglomeration following the "5 + 9 + 6" plan in China.

According to this division principle, the weighted average travel time of each urban agglomeration is shown in Figure 9. In general, the accessibility of national-level urban agglomerations was greater than that of regional-level urban agglomerations, and far greater than that of local-level urban agglomerations. Central Plains urban agglomeration had the shortest weighted average travel time, followed by the Pearl River Delta urban agglomeration and Jianghuai urban agglomeration. Because of the influence of geographical location, terrain, and other restrictive factors, the accessibility of the Chengyu urban agglomeration on a national level was relatively low; thus, other modes of transportation such as the expressway and aviation should contribute to strengthening its connection with the outside population, economy, and industry.

The differences in total economic connection of the 20 urban agglomerations is shown in Figure 10. In general, the total economic connection of national-level urban agglomerations was greater than that of regional-level urban agglomerations, and far greater than that of local-level urban agglomerations. The Pearl River Delta urban agglomeration had the highest total economic connection, followed by the Yangtze River Delta urban agglomeration and Beijing-Tianjin-Hebei urban agglomeration, and the total economic connection of these three urban agglomerations accounted for 46% of all urban agglomerations. The Pearl River Delta urban agglomeration held the highest position in the urban spatial interaction network, with the closest external connection and the strongest external radiation capacity.

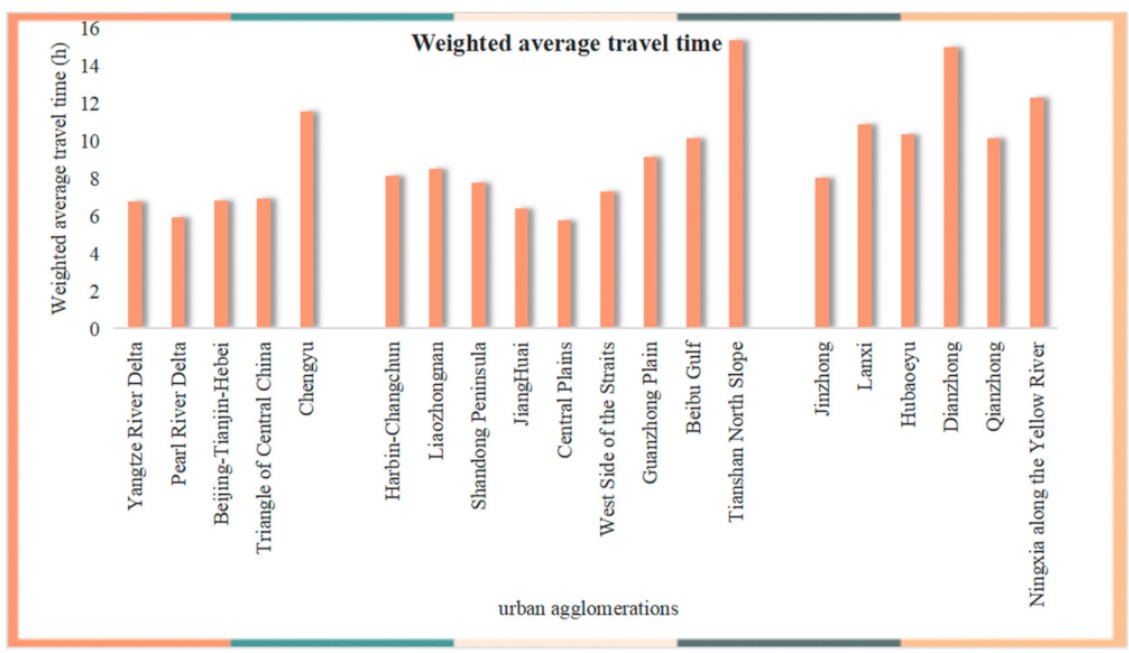

**Figure 9.** Weighted average travel time in 20 urban agglomerations.

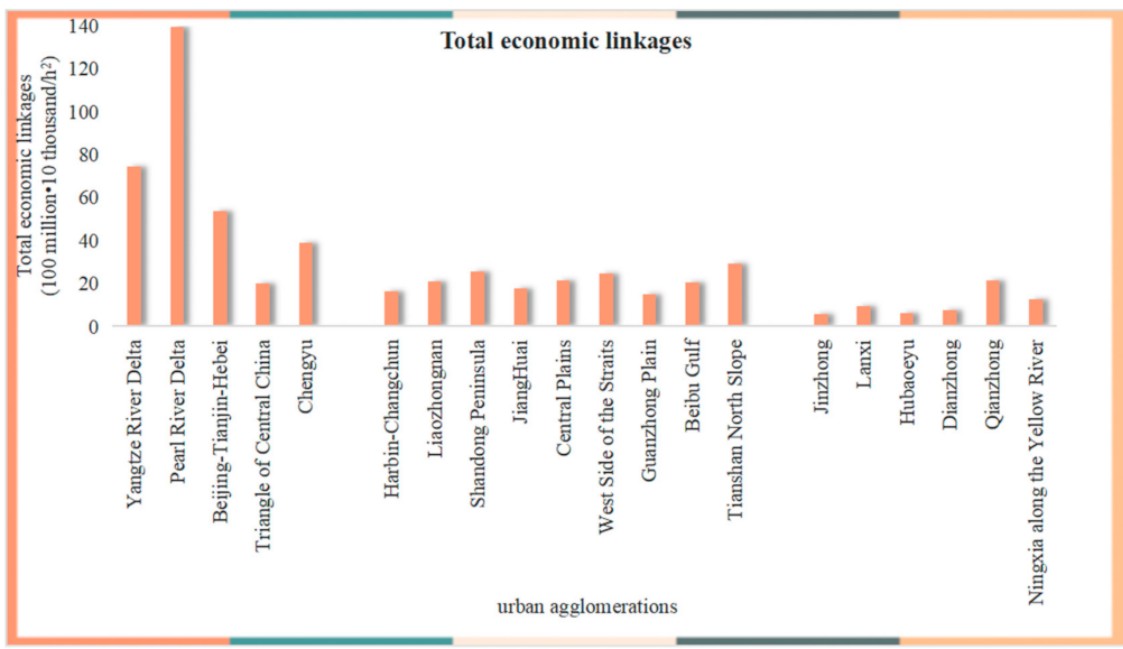

**Figure 10.** Total economic connection in 20 urban agglomerations.

According to the National Comprehensive Three-dimensional Transportation Network Planning document issued on 24 February 2021, the Beijing-Tianjin-Hebei region, Yangtze River Delta, Guangdong-Hong Kong-Macao Greater Bay Area, and Chengyu economic circle are considered to be the main axes; the Triangle of Central China, Shandong Peninsula, West Side of the Straits, Central Plains, Harbin-Changchun, Liaozhongnan, Beibu Gulf, and Guanzhong Plain serve as corridors; Hubaoeyu, Qianzhong, Dianzhong, central Shanxi, Tianshan north slope, Lanxi, Ningxia along the Yellow River, Lhasa and Kashgar are the channels building the main backbone of the national comprehensive three-dimensional transportation network based on the transportation connection intensity [3]. This plan is consistent with the procedures revealed by the accessibility and the total economic

connection of urban agglomerations. The result will better support the national comprehensive three-dimensional transportation network planning and verify the rationality of the planning.

### 4.2. Characteristics of Spatial Connection Pattern and Sustainable Development

The role of transport in sustainable development was first recognized at the 1992 United Nation's Earth Summit and reinforced in its outcome document—Agenda 21. Sustainable transportation can enhance economic growth and improve accessibility. In the 2030 Agenda for Sustainable Development, sustainable transport is mainstreamed across several SDGs and targets, especially those related to food security, health, energy, economic growth, infrastructure, and cities and human settlements [45]. SDG 9.1 recommends developing quality, reliable, sustainable, and resilient infrastructure, including regional and transborder infrastructure, to support economic development and human well-being, with a focus on affordable and equitable access for all. Based on the characteristics of the spatial connection pattern of this study and the sustainable development target 9.1, we analyzed the correlation between the total economic connection and GDP, and the railway passenger volumes and GDP, as shown in Figure 11. The analysis indicated that the total economic connection and railway passenger volume are positively correlated with GDP, with correlation coefficients of 0.74 and 0.67, respectively.

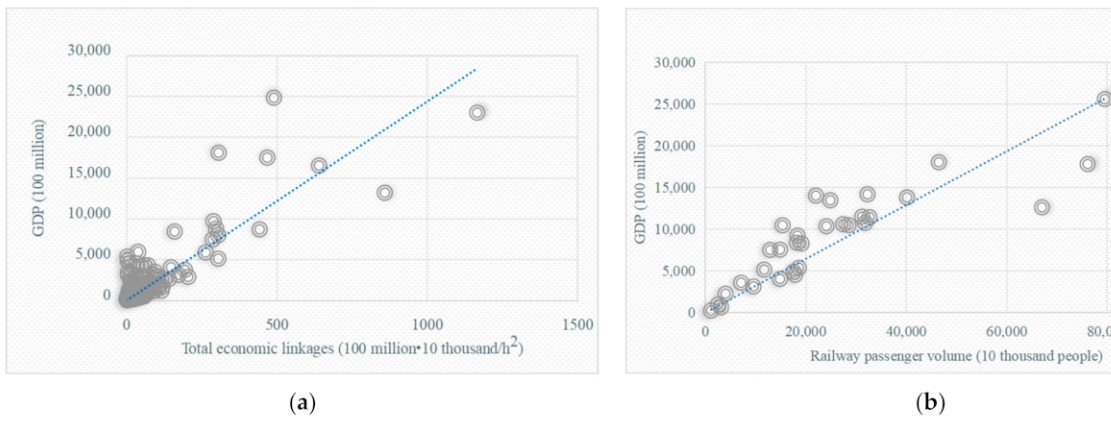

|  (a) | (b) |

**Figure 11.** Characteristics of spatial connection pattern and sustainable development. (**a**) The correlation between total economic connection and GDP; (**b**) The correlation between railway passenger volumes and GDP.

According to the national comprehensive three-dimensional transportation network planning, by 2035 the railway network, which is the backbone of the national comprehensive transportation network, will reach around 200,000 km, and of these, high-speed railway will be 70,000 km and ordinary railway will be 130,000 km. This will effectively promote the more sustainable development of transportation and social economy.

### 4.3. Contributions and Future Improvements

The spatial pattern of the top 50 cities ranked by interurban economic connection based on the big data of railway passenger flow appeared as a "diamond shape". The four vertices of the diamond were the Beijing-Tianjin-Hebei, Yangtze River Delta, Pearl River Delta, and Chengyu urban agglomerations. This diamond pattern was basically consistent with the T-shaped space that has outlined China's regional development for a long time. Relevant research results based on railway passengers, aviation passenger flow, and Tencent migration data have also proved the universality of this law [5,46,47].

The weighted average travel time, railway network density, and economic connection model used in this study are typical methods for describing the spatial connection and pattern and better reflect the characteristics of spatial connection and pattern. This study enriches the "flow space" theory and more fully describes urban network spatial connection

and structure in China by considering the three perspectives of time, space, and interaction. The results showed that there was a mismatch between the layout of the railway network and the population. It will still be necessary to focus on strengthening the construction of transportation infrastructure in urban agglomerations and densely populated areas in the future.

With the increasing maturity of big data (such as RS, GPS, LBS, LSS), new data and methods have started to appear. Because of the high accuracy, large samples of mobile phone data, and the synthesis of various spatial flow information, the study of population flow and spatial connection pattern based on mobile phone data has increasingly become a hot spot that explores the geographical boundaries of human flow, human individual flow mode, travel behavior, and the urban spatial structure [48–51]. In the future, mobile signal and OD data from China Unicom will be used to accurately depict and dynamically simulate the urban spatial connection in China, which can provide a new research perspective and method reference for spatial interaction research.

## 5. Conclusions

Based on the big data of railway passenger flow, the weighted average travel time, railway network density, and the economic connection model were selected to analyze the spatial connection and pattern of China's urban network in 2016 from the three perspectives of time, space, and interaction, respectively, in order to provide guidance and suggestions for the development of national comprehensive three-dimensional transportation network planning, regional spatial structure optimization, and sustainable development. The specific conclusions are as follows:

(1) The accessibility of the railway network presents strip distribution characteristics and reflects significant differences between east and west, and the accessibility of the railway network in the eastern region is higher than that in the western region. The accessibility of national-level urban agglomerations is greater than that of regional-level urban agglomerations, and far greater than that of local-level urban agglomerations.

(2) The railway network density presents regional differences. The railway network density in the eastern region is much greater than that in the western region. Areas with large railway network density are mainly concentrated in the urban agglomerations of Beijing-Tianjin-Hebei, Central Plains, Yangtze River Delta, Jianghuai, and the Triangle of Central China. There is some mismatch between China's railway network layout and population and urban layout. We must strengthen the railway network construction in central and eastern regions with large population densities, especially the Beijing-Tianjin-Hebei, Yangtze River Delta, and other key urban agglomerations.

(3) The total external economic connection in China shows a decreasing trend from the east to the west, as the eastern region is significantly higher than the central region, and much higher than the western region. The total economic connection of national-level urban agglomerations is greater than that of regional-level urban agglomerations, and far greater than that of local-level urban agglomerations. The Pearl River Delta urban agglomeration had the highest total economic connection. The top 50 cities ranked by interurban economic connection showed a "diamond shape" distribution. The four diamond-shaped pairs were Beijing-Tianjin-Hebei, Yangtze River Delta, Pearl River Delta, and Chengyu urban agglomerations.

Traditional research focuses on regional space and urban organizations in the geographical sense, and we should respond positively to the impact of factors such as information technology flow and technological innovation in virtual space. With the emergence of the global urban network, it will be important to explore the vertical connection with the help of various attribute data, and the study of the network structure, function, and relationships among cities and regions at different scales will become a new focus.

**Author Contributions:** Conceptualization, M.L. and B.H.; methodology, X.L. and W.L.; software, B.H.; validation, Y.F.; formal analysis, M.L. and W.G.; investigation, M.L.; resources, M.L.; data curation, M.L.; writing—original draft preparation, M.L.; writing—review and editing, M.L.; visualization, M.L. and W.G.; supervision, W.G. and R.G.; project administration, Z.L.; funding acquisition, M.L. All authors have read and agreed to the published version of the manuscript.

**Funding:** This research was funded by the Humanity and Social Science Youth Foundation of the Ministry of Education of China, grant number 19YJCZH081, and the Open Fund of the Key Laboratory of Urban Land Resources Monitoring and Simulation, Ministry of Natural Resources, grant numbers KF-2020-05-013 and KF-2020-05-035.

**Institutional Review Board Statement:** Not applicable.

**Informed Consent Statement:** Not applicable.

**Data Availability Statement:** Not applicable.

**Acknowledgments:** The authors are grateful for the "Evaluation method and optimization strategy of transportation network connection value based on breaking points model" and "Research on Dynamic Connection of Guangdong-Hong Kong-Macao Greater Bay Area Based on Big Data of Traffic Flow" projects of Shenzhen University. We also would like to thank Jian Sun, who works at Jilin University and gave many suggestions for improving this article.

**Conflicts of Interest:** The authors declare no conflict of interest.

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
