# Peer review of "Urban Network Spatial Connection and Structure in China Based on Railway Passenger Flow Big Data"

_land, doi:10.3390/land11020225_

Round 1

Reviewer 1 Report

The manuscript has been improved though I did not find a response to my earlier comments.

Reviewer 2 Report

I accept the manuscript in this format.

This manuscript is a resubmission of an earlier submission. The following is a list of the peer review reports and author responses from that submission.

Round 1

Reviewer 1 Report

It was a real pleasure to read the manuscript. The authors used large data sets that covered railway passenger flow in the analyzes of urban network spatial connection and structure in China. In the proposed approach, they connected three essential aspects: the "flow space" theory, the spatial interaction model, and the accessibility model. I find it a very interesting and important research topic.

The research method has been described clearly and understandably. Figure 2 presents the framework of the method and is particularly important in this context. In terms of methodology, I have no objections.

The text is very synthetic, consistent, and well structured. The introduction contains a well-described, justified, and properly exposed the goal of the research. The research results were presented in the form of maps and graphs, which greatly facilitates their interpretation and formulation of conclusions.

The content of the manuscript is consistent with its title. The authors also cited many sources from the literature, which I consider very valuable.

Congratulations to the authors for a consistent and interesting approach to the research problem. I recommend publishing the manuscript.

Reviewer 2 Report

dear authors, i have attached a report in word with my comments. 

Reviewer 3 Report

The authors in this paper use railways passenger flow data and the flow space theory in China from 2016 to select weighted average travel time, railway network density, and the economic connection model to express the spatial connection and pattern of China's urban network in 2016 from the perspectives of time, space, and interaction. They also explored the spatial connection pattern of China’s urban agglomerations and the correlation of spatial connection patterns and sustainable development. The weighted average travel time reflects the transportation accessibility, the railway network density reflects the connectivity between regions, and the economic connection model reflects the economic radiation capacity of the central cities to the surrounding areas. The authors conclude that the accessibility of the railway network presents strip distribution characteristics and reflects significant differences between east and west, and the accessibility of the railway network in the eastern region is higher than that in the western region. The railway network density presents regional differences. The railway network density in the eastern region is much greater than that in the western region. The total external economic connection in China shows a decreasing trend from the east to the west, as the eastern region is significantly higher than the central region and much higher than the western region.   This is a useful and well-written paper with contributions to warrant its publication in the journal.   I have two reservations. I am not sure why 2016 is selected for the analysis. Secondly, the description of the proposed methods is shallow and requires further details.  

Reviewer 4 Report

The manuscript is well-written. Its aim is the connection between the density of railway network, the weighted average travel time and the total economic connection of the 20 urban agglomeration.

The three different indicators are not connected adequatety. The average travel time is not expressed the economic linkages of cities.  There are many factors that influence and determine the economic level of a city.

So, the authors of the paper should examine the connections of the examined parameters.